# Prothoracicostatic Activity of the Ecdysis-Regulating Neuropeptide Crustacean Cardioactive Peptide (CCAP) in the Desert Locust

**DOI:** 10.3390/ijms222413465

**Published:** 2021-12-15

**Authors:** Lina Verbakel, Cynthia Lenaerts, Rania Abou El Asrar, Caroline Zandecki, Evert Bruyninckx, Emilie Monjon, Elisabeth Marchal, Jozef Vanden Broeck

**Affiliations:** 1Molecular Developmental Physiology and Signal Transduction, KU Leuven, Naamsestraat 59, P.O. Box 2465, B-3000 Leuven, Belgium; lina.verbakel@kuleuven.be (L.V.); cynthia.lenaerts@hotmail.com (C.L.); rania.abouelasrar@kuleuven.be (R.A.E.A.); caroline.zandecki@kuleuven.be (C.Z.); Evert.Bruyninckx@kuleuven.be (E.B.); emilie.monjon@kuleuven.be (E.M.); Elisabeth.Marchal@imec.be (E.M.); 2Laboratory of Cellular Transport Systems, ON I, Herestraat 49, P.O. Box 802, B-3000 Leuven, Belgium; 3Developmental Neurobiology, KU Leuven, Naamsestraat 61, P.O. Box 2464, B-3000 Leuven, Belgium; 4Imec, Kapeldreef 75, B-3001 Leuven, Belgium

**Keywords:** ecdysis behavior, steroid hormone, neuropeptides, neuroendocrinology, insect, prothoracic gland, receptor

## Abstract

Accurate control of innate behaviors associated with developmental transitions requires functional integration of hormonal and neural signals. Insect molting is regulated by a set of neuropeptides, which trigger periodic pulses in ecdysteroid hormone titers and coordinate shedding of the old cuticle during ecdysis. In the current study, we demonstrate that crustacean cardioactive peptide (CCAP), a structurally conserved neuropeptide described to induce the ecdysis motor program, also exhibits a previously unknown prothoracicostatic activity to regulate ecdysteroid production in the desert locust, *Schistocerca gregaria*. We identified the locust genes encoding the CCAP precursor and three G protein-coupled receptors that are activated by CCAP with EC_50_ values in the (sub)nanomolar range. Spatiotemporal expression profiles of the receptors revealed expression in the prothoracic glands, the endocrine organs where ecdysteroidogenesis occurs. RNAi-mediated knockdown of CCAP precursor or receptors resulted in significantly elevated transcript levels of several *Halloween* genes, which encode ecdysteroid biosynthesis enzymes, and in elevated ecdysteroid levels one day prior to ecdysis. Moreover, prothoracic gland explants exhibited decreased secretion of ecdysteroids in the presence of CCAP. Our results unequivocally identify CCAP as the first prothoracicostatic peptide discovered in a hemimetabolan species and reveal the existence of an intricate interplay between CCAP signaling and ecdysteroidogenesis.

## 1. Introduction

A central tenet in the molting process of arthropods is that this developmental process is initiated by a pulse of ecdysteroid hormones. In juvenile insects, the prothoracic gland (PG) functions as a central node that integrates both intrinsic and extrinsic signals to fine-tune ecdysteroidogenesis and maximize fitness outcomes and developmental progress. During ecdysteroidogenesis, cholesterol is converted to ecdysone (E) and its derivative 20-hydroxyecdysone (20E) by steroidogenic enzymes, many of which belong to the CYP450 family encoded by the *Halloween* genes [1]. Knowledge of the underlying signaling pathways that regulate the onset, duration, and amplitude of ecdysteroid pulses is essential to uncover the complex regulation of the molt, one of the most vital processes within an insect’s life [2].

Prothoracicotropic hormone (PTTH) was the first discovered neurohormone to act as an ecdysteroidogenesis stimulating factor on the prothoracic gland. The release of PTTH controlling developmental timing of metamorphosis via triggered synthesis and secretion of ecdysteroids is a widely accepted paradigm [3]. Additional brain-originated peptides have since been found to aid in the proper regulation of the circulating ecdysteroid titer by exerting either stimulatory (“tropic”) or inhibitory (“static”) effects on the prothoracic gland of holometabolan species. In the domestic silk moth, *Bombyx mori*, peptides reported as stimulators of ecdysteroid synthesis include insulin-like peptides (ILPs), FXPRL-amides, orcokinins and pigment dispersing factor, while the inhibitory factors include prothoracicostatic peptides (PTSPs) or myoinhibtory peptides (MIPs), Bommo-myosuppressin (BMS) and FMRFamide-related peptides (BRFa) [2,3,4,5,6,7,8]. Neuropeptide F (NPF) is, besides PTTH and ILPs, the only confirmed factor to regulate ecdysteroidogenesis in the fruit fly, *Drosophila melanogaster* [9]. Furthermore, several reports in literature suggest that additional peptides may have the potential to regulate prothoracic gland activity. For example, enriched expression of certain neuropeptide receptors, such as the corazonin receptor (CrzR) and the calcitonin-like diuretic hormone receptor 1 (CT/DH-R1), have been demonstrated in the prothoracic gland of the kissing bug, *Rhodnius prolixus* [10,11]. However, whether these signaling systems can (in)directly influence ecdysteroid production or release in the prothoracic gland of *R. prolixus* remains to be investigated.

While molting is typically initiated by a peak in ecdysteroid hormones, the rhythmicity of this process is controlled by neuropeptides. The 20E-induced neuropeptidergic sequence that controls ecdysis has been shown to differ slightly between insect species, despite involving roughly the same neuropeptides [12,13]. In general, more is known about the neuropeptidergic ecdysis sequence in holometabolans than in hemimetabolans. Crustacean cardioactive peptide (CCAP) is one of the central neuropeptides that has been shown to be involved in ecdysis by inducing the ecdysis motor program, the stereotyped sequence of behaviors that mark the end of the pre-ecdysis stage, in several holometabolan insects, such as the tobacco hornworm, *Manduca sexta*, the red flour beetle, *Tribolium castaneum* [14,15], and in the hemimetabolan species *R. prolixus* [16]. 

In the current study, we demonstrate that the CCAP signaling system not only controls the ecdysis motor program in the desert locust, *Schistocerca gregaria*, but also plays a prothoracicostatic role. Mechanistically, this study reveals the existence of a -hitherto unknown- functional interaction between CCAP signaling and ecdysteroidogenesis. The desert locust is one of the most destructive agricultural pests worldwide [17]. In recent years, its huge swarms have severely impacted food security in many countries, ranging from Kenya to India. Mechanistic understanding of pathways controlling crucial processes such as molting may help in identifying novel molecular targets for future insecticide development. Molting is a crucial process in insect development that is intricately regulated by hormones and neuropeptides.

## 2. Results

### 2.1. Fifth Instar Morphology in Correlation to the Ecdysteroid Peak and Structure of the Integument

At each of the sampling moments throughout the fifth nymphal stage (N5), the selected set of physical characteristics (the inter-wing distance, i.e., distance between the wing primordia, and the body weight) and the circulating ecdysteroid levels were measured. As such, it was possible to determine which physical characteristic appeared to be best suited to act as a non-invasive indication of a particular moment in the molt cycle of an individual nymph. Data fitting estimated the occurrence of an ecdysteroid peak at 2.9 ± 0.1 days prior to the N5-Adult (Ad) molt, and correlated this peak with the moment the inter-wing distance reached at least 1.8 mm (N5IW) occurring at 0.2 ± 0.1 days prior to the ecdysteroid peak (Figure 1A,C). In addition, temporal monitoring of the body weight showed a general trend in which the nymphs gained mass daily, which stopped approximately one day prior to ecdysis when their weight decreased with at least 0.1 g (N5WD). Hence, a weight peak (N5WP) for each nymph was estimated at 1.8 ± 0.4 days prior to ecdysis (Figure 1B,C). This weight peak occurred on average 1.0 ± 0.4 days after the peak in the circulating ecdysteroid titer.

To further confirm whether these physical indicators are representative for certain milestones in the N5 nymph’s molt cycle, changes in the histology of the integument at these sampling moments (N5IW, N5WP and N5WD) were evaluated. The width of the exuvial space was observed to increase significantly and gradually at these different sampling moments. Moreover, the old endocuticle significantly decreased in width at N5WD, compared to the other sampling points (Figure 1D).

### 2.2. Identification, Molecular and Functional Characterization of the S. gregaria CCAP Precursor and Receptors

The *Schgr*-CCAPR-1, -2 and -3 ORF sequences consist of 1236, 1206 and 1416 nucleotides encoding polypeptides of 412, 402 and 472 amino acid residues, respectively.

Transmembrane helix (TM) topology prediction revealed the presence of seven hydrophobic regions forming the α-helical TM (1–7) segments that are characteristic for GPCRs [18]. A multiple sequence alignment uncovered similarities of these cloned receptors with cognate CCAPRs (Appendix A). They display many typical features of rhodopsin-like receptors, such as the DRY motif in the second intracellular loop (IL2) near TM3 and two conserved cysteines that are predicted to form a disulfide bridge connecting the first and second extracellular loops (EL1–EL2), thereby ensuring a proper 3D orientation, which is believed to be important for ligand binding [19]. In addition, two, three and no putative N-glycosylation sites, according to the NXS/T consensus sequence, are observed in the extracellular N-terminal region of *Schgr*-CCAPR-1, -2 and -3, respectively. In addition, a maximum-likelihood phylogenetic analysis of CCAPR sequences for the selected species showed that the three *Schgr*-CCAPRs are most closely related to the orthologous sequences of the band-legged ground cricket, *Dianemobius nigrofaciatus*, the gray bird grasshopper, *Schistocerca nitens*, other orthoptheran species, followed by species in other polyneopteran orders, such as the drywood termite, *Cryptotermes secundus* (Blattodea), and the dampwood termite, *Zootermopsis nevadensis* (Isoptera) (Appendix A).

The *Schgr*-CCAPpre ORF sequence is composed of 447 nucleotides and codes for a prepropeptide of 149 amino acids, containing a signal peptide of 24 amino acids at the N-terminus, predicted by SignalP-5.0 [20], and several dibasic, tribasic and/or RxxR putative cleavage sites. This polypeptide contains one single copy of the highly conserved neuropeptide CCAP from amino acid position 50 until 59, which is followed by an amidation signal as expected (Appendix A).

To deorphanize the three GPCRs, they were heterologously expressed in Chinese hamster ovary (CHO)-WTA11 cells, which stably express aequorin and a promiscuous Gα16 subunit. CCAP elicited dose-dependent aequorin-based bioluminescent responses in CHO-WTA11 cells expressing *Schgr*-CCAPRs, with half-maximal effective concentration (EC_50_) values in the (sub)nanomolar range: 3.65 nM, 0.32 nM and 0.59 nM for *Schgr*-CCAPR-1, -2 and -3, respectively (Figure 2A). Intracellular signaling of *Schgr*-CCAPR-1, -2 and -3 was investigated in *Schgr*-CCAPR-expressing cells lacking the promiscuous Gα16 subunit (CHO-PAM28 cells) or co-expressing a cyclic AMP (cAMP) response element (CRE)-luciferase reporter (human embryonic kidney (HEK) 293T). In *Schgr*-CCAPR expressing CHO-PAM28 cells, *Schgr*-CCAP induced dose-dependent aequorin-based bioluminescent responses, with EC50 values in the subnanomolar range: 0.7 nM, 0.2 nM and 0.5 nM for *Schgr*-CCAPR-1, -2 and -3, respectively (Appendix A). In addition, when exposed to CCAP, *Schgr*-CCAPR expressing HEK-293T cells, co-transfected with a CRE-controlled luciferase reporter gene, were able to dose-dependently elevate luciferase activity levels, with EC_50_ values in the (sub)nanomolar range: 0.01 nM, 1.2 nM and 0.05 nM for *Schgr*-CCAPR-1, -2 and -3, respectively (Appendix A).

To verify the agonist selectivity of the three *Schgr*-CCAPRs, several other *S. gregaria* peptides (Appendix A) were tested as well. At micromolar concentrations (1 µM), only CCAP markedly activated the *Schgr*-CCAPRs in CHO-WTA11 cells (Figure 2B).

No responses were observed when the CHO-WTA11, CHO-PAM28 and HEK293T cells, transfected with an empty pcDNATM3.1 vector (in addition to CRE_(6x)_-*luc* reporter construct in case of HEK293T), were exposed to 1 µM of the tested *S. gregaria* peptides (Appendix A) or to BSA-only medium (negative controls).

### 2.3. Localization and Developmental Expression Patterns of the S. gregaria CCAP Precursor and Receptors

Analysis of the spatial expression pattern showed that *Schgr*-CCAPpre is expressed in CNS-derived tissues (Appendix A).

The tissue distribution of the three *Schgr*-CCAPR transcripts was also determined at N5WD. The *Schgr*-CCAPR-1 transcript was predominantly observed in different parts of the CNS and in several more peripheral tissues, such as prothoracic gland, corpora cardiaca, trachea and hindgut (Appendix A). In comparison with *Schgr*-CCAPR-1, *Schgr*-CCAPR-2 was more widely expressed, showing measurable expression levels in the prothoracic gland, corpora cardiaca, corpora allata, trachea, fat body and flight muscle (Appendix A). *Schgr*-CCAPR-3 also had a wide tissue distribution, with detectable expression levels in most parts of the CNS and in peripheral tissues, such as prothoracic gland, corpora cardiaca, corpora allata and fat body (Appendix A).

Considering the observed expression of all three *Schgr*-CCAPR transcripts in the prothoracic gland, the temporal expression profiles of the *Schgr*-CCAPRs were analyzed in the prothoracic gland of N5 nymphs. While *Schgr*-CCAPR-1 and -3 transcript levels seemed to abruptly increase at the molt cycle moment N5WD by approximately 7 and 5-fold, respectively (Figure 3A), the transcript levels of *Schgr*-CCAPR-2 increased more gradually during the second half of N5. Interestingly, the *Schgr*-CCAPR-2 transcript levels started to increase at the sampling moment nearest to the ecdysteroid peak in N5 (N5IW, Figure 3A).

### 2.4. Knockdown of Ecdysteroid Receptor Components Influences Expression of Schgr-CCAPR Transcripts in the Prothoracic Gland

To further investigate whether *Schgr*-CCAPR expression in the prothoracic gland might be regulated by components of the heterodimeric ecdysteroid receptor complex, the *Schgr*-EcR and *Schgr*-RXR transcripts were reduced by RNAi to 14% and 39%, respectively (Appendix A, respectively). The *Schgr*-CCAPR-1, -2, and -3 transcript levels at N5WD were significantly decreased by 70.6%, 97.8% and 44.5%, respectively (Figure 3B), compared to *dsGFP*-injected (control) nymphs.

### 2.5. Phenotypic and Molecular Effects of CCAP Precursor or Receptor RNAi-Mediated Knockdown

#### 2.5.1. Knockdown Efficiency of the dsCCAP(pre/R) Constructs

As a consequence of *dsCCAPpre*-injection, only the transcript levels of *Schgr*-CCAPpre were dramatically reduced by 99% (Appendix A), while no significant changes were observed in the transcript levels of *Schgr*-CCAPR-1 and -3 (Appendix A, respectively). *dsCCAPR1/2/3a*-injections effectively reduced the *Schgr*-CCAPR-1, -2 and -3 transcript levels to 34%, 15% and 48%, respectively (Appendix A, respectively). Similarly, injections of *dsCCAPR1/2/3b* caused a significant reduction in the levels of *Schgr*-CCAPR- 1, -2 and -3 transcripts to 27%, 28% and 27%, respectively (Appendix A, respectively). Furthermore, the construct *dsCCAPR-1* induced a reduction of the *Schgr*-CCAPR-1 transcript levels to 15% (Appendix A), while leaving transcript levels of *Schgr*-CCAPR-2 and -3 unaffected (Appendix A, respectively). Likewise, the constructs *dsCCAPR-2* and *-3* selectively reduced the *Schgr*-CCAPR-2 and -3 transcript levels to 20% and 17%, respectively (Appendix A, respectively).

#### 2.5.2. RNAi Ecdysis Phenotypes

Locusts belonging to the different experimental groups did not display phenotypic differences until the expected time of ecdysis. *dsCCAPpre-*, *dsCCAPR1/2/3a*- and *dsCCAPR1/2/3b*-injections, starting from fourth instar nymphs, resulted in, respectively, 93%, 87% and 91% of the locusts in an ecdysis deficient phenotype (Figure 4A). Moreover, injections that started from late third instar nymphs, with these three constructs, led to 100% ecdysis deficiency, considering the combined effect on the N4–N5 molt and the N5-Ad molt in the tested population (Figure 4B). Furthermore, insects with these transcripts knocked down exhibited no significant delay in ecdysis initiation (Figure 4C).

Knockdown animals with defective ecdysis showed tense hind- and forewing primordia standing straight up with a wide separation between these primordia (Figure 4E top). Moreover, these nymphs were continuously displaying pumping behavior in an attempt to shed their old cuticle. Manual removal of the old larval cuticle revealed that the new adult cuticle was already formed when the nymphs were arrested at ecdysis (Figure 4E bottom). Together this indicated that they died after initiating pre-ecdysis behavior. By contrast, nymphs injected with *dsGFP* (control) or *dsCCAPR-1*-, *-2*- or *-3* (targeting highly diverging regions of the respective receptors) managed to shed their old cuticle (Figure 4A,B). In *Schgr*-CCAPpre depleted nymphs, a significant rescue effect (60% (18/30)) of the ecdysis deficiency phenotype was obtained by daily injecting 0.92 pg CCAP peptide throughout N5, when compared to a group of nymphs that were daily injected with *S. gregaria* saline only (Figure 4D).

Since a selective knockdown of *Schgr*-CCAPR-1, -2 and -3 did not result in ecdysis deficits (Figure 4A), we tried to test if combined injections of the *dsCCAPR-1*, *-2* and *-3* constructs would result in ecdysis deficiency. 86% of the nymphs that were injected with all three selective *dsCCAPR* constructs displayed ecdysis deficits, whereas 0% of the nymphs injected with *dsGFP* (control) showed ecdysis defects (Figure 4F). The combinatorial treatments with *dsCCAPR-1* + *dsCCAPR-2* constructs and *dsCCAPR-2* + *dsCCAPR-3* constructs resulted in, respectively, 31 and 15% of the injected nymphs with a clear ecdysis deficiency, while nymphs that received *dsCCAPR-1* + *dsCCAPR-3* constructs did not exhibit defective ecdysis (Figure 4F).

#### 2.5.3. Effects on Halloween Gene Expression in the Prothoracic Glands

Both *dsCCAPpre-* and *dsCCAPR1/2/3a*-injections resulted in significantly elevated levels of the *Halloween* gene transcripts coding for *Schgr*-Spook (95% and 91%, respectively), *Schgr*-Disembodied (43% and 53%, respectively) and *Schgr*-Shade (95% and 91%, respectively) at N5WD, when compared to *dsGFP*-injected (control) nymphs (Figure 5A). Moreover, the ecdysteroid levels were also significantly increased in the hemolymph of these knockdown nymphs at N5WD (Figure 5B). Similar results were observed at N5WD in *dsCCAPR1/2/3b*-injected nymphs (Figure 5A). However, no changes in the levels of the *Halloween* transcripts or circulating ecdysteroids were detected in *dsCCAPR-1-* or *dsCCAPR-3*-injected nymphs at N5WD (Figure 5A,B, respectively). Despite a significant elevation of *Schgr*-Spook transcript levels in *dsCCAPR-2-*injected nymphs (Figure 5A), no significant effect was detected on the levels of other *Halloween* gene transcripts (Figure 5A), nor on ecdysteroid levels (Figure 5B). In addition, no significant shift in the *Halloween* gene transcript levels was observed at earlier time points within the molt cycle (N5IW and N5WP) in *dsCCAPpre*- or *dsCCAPR1/2/3a*-injected nymphs (Appendix A).

### 2.6. Effect of CCAP on Ecdysteroidogenesis in the Prothoracic Glands

In order to investigate whether CCAP acts as a direct mediator on the prothoracic gland to influence ecdysteroidogenesis, prothoracic gland explants micro-dissected at N5IW were incubated with or without CCAP and the secreted ecdysteroid levels were subsequently quantified by EIA. CCAP exhibited a significant prothoracicostatic activity on these prothoracic gland explants (Figure 6).

## 3. Discussion

### 3.1. Physical Characteristics Can Be Used to Identify Distinct Points within the Molt Cycle

The molt cycle is a very critical period in the development of a juvenile insect. It coordinates larval or nymphal development and is tightly regulated by the insect’s nervous and endocrine systems [12,21]. Reports in multiple arthropod species established that surges of 20E occur in each instar committing the animal to the molt [21]. Molting individuals stop feeding, which leads, as observed, to a weight decrease prior to the ecdysis process. Once initiated, the commitment to the molting process is irreversible. Nevertheless, various environmental factors, such as food availability, infection, wounding, and temperature can affect how long it takes for an individual to initiate the molting process in each instar. To standardize distinct moments more accurately in the molt cycle of each individual nymph in the studied population, physical characteristics of *S. gregaria*, were determined correlating to the ecdysteroid titer and preceding ecdysis by approximately three, two and one day(s), respectively: N5IW, N5WP and N5WD. Moreover, based on microscopic analysis of the integument, these characteristics can also be used to indicate distinctive morphological phases in the molt cycle:The observed rise in interwing distance (IW) correlates with the onset of the molting cycle, approximately marking the peak in circulating ecdysteroids and a significant increase in exuvial space compared to day 4 in the fifth nymphal stage (N5D4).The weight peak (WP) coincides with a noticeable and significant increase in the exuvial space compared to N5IW.The weight decrease (WD) is associated with an additional noticeable increase in exuvial space compared to N5WP, as well as significant shrinkage of the old endocuticle.

Nymphs of the migratory locust, *Locusta migratoria,* have also been reported to exhibit a WP approximately 48 h before the onset of ecdysis. Their study also demonstrated that the absolute daily mass gain is a function of instar number, body size, and temperature [22]. Hence, these identified molt-related visible features were used to developmentally synchronize individual nymphs in their molt cycle determining sampling moments for dissections and experimental analyses.

### 3.2. The Locust’s CCAP Signaling System Is Structurally and Functionally Conserved

In agreement with reports from other insects, the general organization of the *Schgr*-CCAP prepropeptide is well conserved [23,24,25]. It contains a single copy of the CCAP neuropeptide sequence, as well as another region that is evolutionarily preserved in the CCAP prepropeptide sequences of insects.

The three *Schgr*-CCAPRs show high sequence similarity to deorphanized CCAP receptors of *D. melanogaster*, *T. castaneum* and *R. prolixus*. Although only one CCAP receptor has been identified in many insect species, there are other species, such as *S. gregaria*, which have multiple isoforms [26,27,28,29,30]. Phylogeny suggested frequent and rather recent independent duplications of CCAP receptor genes, as also concluded by earlier CCAP receptor phylogenetics [26,28]. This is a commonly occurring feature during evolution of GPCR genes in both vertebrates and invertebrates [31]. To support our phylogenetic analysis, which suggests that the identified receptors are indeed homologous sequences of previously characterized CCAP receptors, we tested these putative *Schgr*-CCAPRs in cell-based receptor activity assays and concluded that CCAP is a very potent, naturally occurring agonist of the *Schgr-*CCAPRs. Further investigation of intracellular signaling hinted that the *Schgr-*CCAPRs possibly display dual coupling to both cAMP and Ca^2+^ second messenger systems upon activation. A stimulatory effect on intracellular Ca^2+^ levels was previously also suggested for the characterized CCAPRs of *T. castaneum* (*Trica*-CCAPR-1 and -2) [26]. Moreover, previously reported in-vivo data also demonstrated that CCAP induces Ca^2+^-dependent contractions in locust oviducts [32]. However, to our knowledge, effects on cAMP response element (CRE)-dependent reporter expression had not been reported for CCAP so far.

### 3.3. The Imperative Role of the CCAP Signaling System in Ecdysis Is Conserved in S. gregaria

The three receptor genes are broadly expressed in a variety of tissues. Likewise, the CCAPR of *R. prolixus* [28] was also detected in a number of peripheral tissues. These broad *Schgr*-CCAPR-1, -2 and -3 expression profiles suggest regulatory roles for CCAP in central and peripheral physiological processes. Ecdysis of nymphs injected with *dsCCAPpre*, *dsCCAPR1/2/3a* or *-b* constructs were clearly arrested after the pre-ecdysis process was completed. Moreover, this ecdysis-deficient phenotype was partially rescued by injections of the CCAP peptide. The key role of the CCAP signaling system in the initiation of ecdysis behavior has been proven in several holometabolan species [14,15,33,34], as well as in the hemimetabolan species *R. prolixus* [16]. The current study shows that CCAP is also a crucial factor for ecdysis in *S. gregaria*, a major pest species. Hence, this points towards a conserved role of the CCAP signaling system in ecdysis. The fact that a knockdown of all three *Schgr-*CCAPR transcripts resulted in an ecdysis-deficient phenotype, while combinations of only two selective *dsCCAPR* constructs yielded weaker phenotypes and locusts injected with a single gene-specific construct (*dsCCAPR-1, -2* or *-3*) appeared to undergo a normal ecdysis, hints at a functional redundancy of the three *Schgr*-CCAPRs in the neuropeptidergic cascade regulating ecdysis in *S. gregaria*. By contrast, Li et al. (2011) demonstrated receptor sub-functionalization of *Trica*-CCAPR-2 within the ecdysis process of *T. castaneum* [26]. 

### 3.4. Identification of a Prothoracicostatic Function of the CCAP Signaling System in S. gregaria

Nymphs with a disrupted CCAP signaling system displayed elevated ecdysteroid levels as well as significantly increased expression of the *Halloween* genes, *Schgr-Spook*, *Schgr-Disembodied* and *Schgr-Shade* at N5WD. High relative transcript levels of the *Schgr*-CCAPRs were determined in the PG at N5WD. Moreover, *Schgr*-CCAP suppressed ecdysteroid secretion by prothoracic gland explants derived from N5IW nymphs. From these observations we conclude that CCAP acts directly on the prothoracic gland as a prothoracicostatic factor. In contrast to the ecdysiostatic activity of CCAP in *S. gregaria* nymphs, daily injection of CCAP in another hemimetabolan insect, the adult American cockroach, *Periplaneta americana*, resulted in elevated 20E levels [35]. These opposed observations might be due to differences in regulation at another developmental stage and/or between species.

We show significantly reduced transcript levels of the *Schgr*-CCAPRs in *Schgr-EcR/Schgr-RXR* depleted nymphs and developmental timing dependent expression of *Schgr*-CCAPRs in the prothoracic gland. Notably, in both holometabolan insects and crustaceans, CCAP is released in significant quantities into the hemolymph during ecdysis, acting as a neurohormone to stimulate heartbeat and contraction of skeletal muscles [14,15,33]. Since our results demonstrate that CCAP is pivotal for the initiation of the ecdysis motor program in this species, it is likely that CCAP release follows a similar pattern during ecdysis of *S. gregaria*. We hypothesized, therefore, that it is unlikely that the CCAP signaling system is responsible for the initial decline of the circulating ecdysteroid levels initiating the onset of ecdysis. We have provided further evidence for this hypothesis as no changes in *Halloween* gene transcript levels were detected in *Schgr*-CCAPpre- or *Schgr*-CCAPR-1, -2, -3 depleted nymphs at earlier timepoints during the molting cycle. CCAP likely plays a more prominent role in further suppression of prothoracic gland activity to fine-tune ecdysteroid synthesis, while it may also have effects on ecdysteroid release by the prothoracic gland [2].

We further investigated the prothoracicostatic activity of the CCAP signaling system downstream of CCAP through functional characterization of the three *Schgr*-CCAPRs. Elevated levels of *Schgr-Spook* were observed in prothoracic glands of *Schgr*-CCAPR-2 depleted N5WD nymphs. However, this did not seem to be sufficient to significantly elevate the circulating ecdysteroid levels. Hence, even though *Schgr*-CCAPR-2 may play an important role in the direct prothoracicostatic activity of CCAP, the lack of shifts in circulating ecdysteroid and *Halloween* transcript levels upon selective (single-transcript) *Schgr*-CCAPR knockdown suggest that receptor redundancy perhaps occurs as a potential compensatory mechanism.

## 4. Materials and Methods

### 4.1. Insect Rearing

The *S. gregaria* locusts were reared under crowded conditions as previously described [36]. In the described experiments, N5 hoppers were collected within 24 h after ecdysis to obtain cohorts of more synchronized animals.

### 4.2. Sampling Points Based on Locust Morphology

Female N5 nymphs (*n* = 25) were synchronized on the day of ecdysis (N5 day 0, N5D0) and two temporally and visibly changing physical characteristics, body weight and inter-wing distance (i.e., the distance between the wing primordia), were measured at N5D0. These two physical characteristics were then systematically evaluated daily as of day 4 (N5D4) until the adult molt. These anatomical indicators were selected based on their applicability in experimental set-ups.

### 4.3. 20E Enzyme Immunoassay

To correlate the locust’s morphology to the nymphal ecdysteroid titer that regulates molting, hemolymph was collected as well at the described sampling moments. 20 µL hemolymph from each N5 animal was collected into 280 µL cold ethanol. Each sample was further processed as previously described [37]. The ecdysteroid titers were subsequently quantified using the 20-Hydroxyecdysone ELISA kit (Bertin Technologies, Montigny-le-Bretonneux, France), according to the manufacturer’s protocol.

### 4.4. Correlating Locust Morphology to Ecdysteroid Titer

The data were fitted to verify a correlation between the developmental timing of the peak in 20E titer and the temporal changes that occur in both physical characteristics. Such a correlation was suggested when these temporal data were plotted in function of days prior to ecdysis. The developmental time points of the WP, the 20E peak and sigmoid midpoint of the inter-wing distance were extracted from fitted functions.

The weight increase data were fitted as a Lognorm-function:$$f\left(x;µ,\sigma \right)=\frac{A}{\sigma \sqrt{2\pi }}\;\frac{e^{-\frac{{\left(\ln\left(x\right)-µ\right)}^{2}}{2\sigma^{2}}}}{x}$$

The inter-wing distance followed a linear pattern until a critical point, after which it stayed constant. Therefore, it was fitted by a linear and constant function that are linked with a logistic function:$$f\left(x;A,µ,\sigma ,form{=}^{\prime }logistic^{\prime }\right)=A\left(1-\frac{1}{1+e^{\alpha }\;}\right);a=\frac{\left(x-µ\right)}{\sigma }$$

Note that *σ*, which defines the width of the link-regime, was fixed such that the intersection width has a reasonable width compared to experimental observation of the inter-wing distance. The fit result µ (the sigmoid midpoint) was defined as the time point where “the shift” between linear and constant inter-wing distance occurs.

The ecdysteroid data were fitted as a Lorentzian function:(1)$$f\left(x;A,\;µ,\;\sigma \right)=\frac{A}{\pi }\;\left(\;\frac{\sigma }{{\left(x-µ\right)}^{2}+\sigma^{2}}\right)$$

The uncertainty of the calculated developmental time points was determined by error propagation considering the uncertainties of the fit parameters. Correlations were determined by subtraction of the estimated µ values, and the uncertainties of the individual developmental time points were considered.

### 4.5. Histology

The integument of female nymphs was collected at N5D4, N5IW, N5WP and N5WD from the first dorsal abdominal segment and fixed for 7 days at 4 °C in 4% paraformaldehyde (PFA) in phosphate-buffered saline (PBS, pH = 7.4; Sigma-Aldrich, St. Louis, MO, USA). Fixed abdominal segments were rinsed in PBS and subsequently transferred to a solution of 0.9% NaCl, followed by progressive dehydration induced by an ethanol gradient (50–70–80–90–100%). Next, dehydrated abdominal segment tissues were incubated in a xylene:ethanol (1:1) mixture, followed by incubation in a 100% xylene solution and paraffin embedding, after which 6 µm sections were cut using a paraffin microtome. Paraffin sections were mounted on glass slides for histology. Therefore, paraffin-embedded integument sections were deparaffinized and progressively rehydrated by incubating them consecutively in xylene, 100–70–50% and Milli-Q (MQ) water. Subsequently, rehydrated paraffin-embedded integument sections were stained with hematoxylin and eosin (H&E). Excess dye was removed by thoroughly rinsing the slides with running tap water. Next, H&E stained sections were again dehydrated by a graded ethanol series and xylene and mounted in Mowiol mounting medium (Sigma-Aldrich, St. Louis, MO, USA) on glass slides. For morphological analysis of the integument, all stained sections were imaged using an AxioCam MRc5 camera connected to a Zeiss Imager.Z1 microscope (Zeiss, Oberkochen, Germany) and the Imaging software program Zen 2012 (Zeiss, Oberkochen, Germany). The image processing software Image J was used for the analysis of the integument layer thickness as well as of the exuvial space width. To reduce bias and orientation-determined variability these analyses were conducted on at least three sections per nymph (technical replicates) in which at least 5 different points were measured. Furthermore, at each morphology-based point in the molt cycle, the dorsal abdominal segment integument of 5 different N5 nymphs (biological replicates) was analyzed. All statistical tests were performed using GraphPad Prism 6.01 (Graphpad Software, San Diego, CA, USA. In all cases, raw data were tested for normal distribution using the Kolmogorov-Smirnov normality test and variance between groups was checked via the Brown–Forsythe’s test for equality of variances. To evaluate differences between the width of endocuticle or of the exuvial spaces at different molt cycle moments, a one-way ANOVA was performed, followed by a Dunnett’s test.

### 4.6. Cell-Based Reporter Assays

Aequorin and CRE_(6x)_-*luc* reporter assays, as well as maintenance of the cell cultures and cell transfections, were carried out as previously described [36]. Ligand-induced bioluminescent responses were normalized to the maximal 100% response level, which was observed at 1 µM *Schgr*-CCAP in the receptor expressing cells. Minimum bioluminescence values correlate with the signal obtained in the blank condition (without exposure to *S. gregaria* peptides). Calculations were performed using the output file from the Microwin software (Berthold Technologies, Bad Wildbad, Germany) in Excel (Microsoft). Half-maximal effective concentrations (EC_50_) were determined with the PRISM software (Graphpad version 6.03) using non-linear regression, log(agonist) vs. normalized response, applied to a sigmoidal dose-response model.

### 4.7. Cloning, Sequence Analysis and Phylogeny of Schgr-CCAPpre and Schgr-CCAPRs

The complete ORFs of *Schgr-*CCAPpre and of three putative *Schgr-*CCAPRs, *Schgr*-CCAPR-1, -2 and -3, were acquired by tBLASTn analysis using the most recently curated *S. gregaria* transcriptome and genome data [38] (transcript SCHGR_00011839, SCHGR_00011645, SCHGR_00014680 and SCHGR_0001660 from locus 100982-225313, 103086-401037, 2999-34131 and 758-21576 of the seq_7459, seq_3541, seq_9570 and seq_9032 contig, respectively). Therefore, the amino acid sequence of *B. mori* CCAPpre (NM_00130897.1) was used as a query for finding *Schgr-*CCAPpre, while the amino acid sequence of *D. melanogaster* CCAPR (NM_206574.4) was used as query to identify the *Schgr*-CCAPRs.

Primers specific to the hit regions are depicted in Appendix A, and were designed and used to amplify cDNA sequences encoding these complete ORFs by nested PCR in a Doppio Thermal Cycler (VWR, Radnor, PA, USA), using Q5^®^ High-Fidelity DNA Polymerase (New England Biolabs, Ipswich, MA, USA). The cDNA template was synthesized from total RNA that was extracted from thoracic ganglia, using the Transcriptor High Fidelity cDNA Synthesis Kit (Roche, Basel, Switzerland). After gel electrophoresis on a 1% agarose gel, UV was used to visualize the resulting PCR amplicons that were further extracted using the GenElute Gel Extraction Kit (Sigma-Aldrich, St. Louis, MO, USA). Next, they were cloned into a pJET2.1/blunt vector (Thermo Fisher Scientific, Waltham, MA, USA) and transformed into One Shot TOP10 chemically competent *E. coli* cells (Invitrogen, Waltham, MA, USA). Transformed bacterial colonies were then grown overnight at 37 °C on LB agar plates supplemented with 100 ng/mL Ampicillin (Invitrogen, Waltham, MA, USA). Individual colonies were verified by means of colony PCR and positive colonies were grown overnight at 37 °C in LB medium supplemented with 100 ng/mL Ampicillin (Invitrogen, Waltham, MA, USA). Subsequently, cloned plasmids were isolated using the GenElute Plasmid Miniprep kit (Sigma-Aldrich, St. Louis, MO, USA) and the sequence of their inserts was verified by Sanger sequencing. Finally, confirmed receptor cDNA sequences were subcloned into a pcDNA^TM^ 3.1/V5-His-TOPO^TM^ TA expression vector (Invitrogen, Waltham, MA, USA).

The amino acid sequences of the *Schgr*-CCAPRs and these of other insect CCAP receptors (*R. prolixus*, *Rhopr*-CCAPR, AGT02811.1; *D. melanogaster*, *Drome*-CCAPR, AAO66429; *T. castaneum*, *Trica*-CCAPR-1 and -2, ABN79651 and ABN79652, respectively) were subjected to a multiple sequence alignment. This was also done with the amino acid sequence of the *Schgr*-CCAPpre and these of other CCAP precursors from different species (*R. prolixus*, *Rhopr*-CCAPpre, ADK73625.1; *D. melanogaster*, *Drome-*CCAPpre, AGB96226.1; *Manduca sexta*, *Manse-*CCAPpre, AAL39064.1; *B. mori*, *Bommo-*CCAPpre, BAG50376.1). The EMB-EBI Clustal Omega Multiple Sequence alignment software (http://www.ebi.ac.Tools/msa/clustalo/) was used to create a multiple sequence alignment of the three *Schgr-*CCAPRs. Figures of the alignments were created by Boxshade. Putative transmembrane regions (TM) were predicted by the TMHMM Server v.2.0 (http://www.cbs.dtu.dk/services/TMHMM/). The N-glycosylation sites were predicted using server NeyGlyc 1.0 (http://www.cbs.dtu. dk/services/NetNGlyc/). The signal peptide was predicted by SignalP-5.0. Phylogenetic analysis of the aligned sequences and additional (putative) non-characterized CCAP receptor sequences (*Schistocerca nitens*, *Schni*-CCAPR-1 and -2, GIOW01102367.1, GIOW01112628.1, respectively; *Dianemobius nigrofasciatus*, *Diani*-CCAPR-1 and -2, IADE01117080.1 and IADE01117083.1, respectively; *Cryptotermes secundus*, *Cryse*-CCAPR, XM_033755399.1; *Zootermopsis nevadensis*, *Zoone*-CCAPR, XP_021921974.1; *Aphis gossypii*, *Aphgo*-CCAPR, XM_027997733.1; *Cimex lectularius*, *Cimle*-CCAPR-1 and -2; XP_014248905.2 and XP_014248903.1, respectively; *B. mori*, *Bommo-*CCAPR-1 and -2, NP_001127724.1 and NP_001127746.1, respectively; *Drosophila virilis*, *Drovi*-CCAPR, GJ23325; *Drosophila mojavensis*, *Dromo*-CCAPR, GI22912; *Aedes aegypti*, *Aedae*-CCAPR-1 and -2, AGAP001961 and XP_321100.4, and *Culex pipiens*, *Culpi*-CCAPR-1 and -2, CPIJ006268 and XP_001847670.1) were used for a maximum likelihood analysis with IQ-tree (http://iqtree.cibiv.univie.ac.at/, 1000 SH-aLRT replicates and 10000 ultrafast bootstrap replicates, [39]). The *S. gregaria* adipokinetic hormone receptor (AKHR, AVG47955.1) was included in the analysis as outgroup. iTOL (https://itol.embl.de/) was used to visualize the annotated phylogeny. The online tools mentioned above were accessed between December 2020 and September 2021. 

### 4.8. RNA Interference Experiments

The primers used to generate the dsRNA constructs are listed in Appendix A. The *dsRNA* constructs were designed to target *Schgr-CCAPpre* or either highly similar or highly diverging sequence regions of *Schgr*-CCAPR-1, -2 and -3. In particular, the *dsCCAPpre* construct targets *Schgr-CCAPpre*, the *dsCCAPR1/2/3a* and *dsCCAPR1/2/3b* constructs target highly conserved regions shared by *Schgr*-CCAPR-1, -2 and -3, and the selective *dsCCAPR-1*, *dsCCAPR-2* and *dsCCAPR-3* constructs target highly diverging sequence regions of *Schgr-CCAPR-1*, *-2* and *-3*, respectively. The dsRNA constructs for *Schgr-EcR* and *Schgr-RXR* were previously characterized [40]. The injected *dsRNA* quantities were based on pilot experiments, in which the different *dsRNAs* were tested, since the optimal dose may vary depending on the targeted gene transcript.

***Schgr-CCAP(pre/R)* knockdown**. Locusts were injected into the hemocoel with 4 µL (100 ng/µL) of *dsCCAPpre/R* constructs within 24 h after their N3-N4 molt. Boost injections were administered at N4D4, N5D1, N5D4 and N5D7. Control locusts were injected with an equal concentration of *dsGFP*. Locusts (*n* = 15) were dissected into 5 pools per condition at N5WD to investigate effects of RNAi-mediated depletion. In addition, 25 locusts were used for observance of ecdysis deficits.

***Schgr-EcR/Schgr-RXR* knockdown.** Locusts were injected into the hemocoel with 4 µL *dsEcR/RXR* (50 ng/µL) using the constructs and injection schemes, as previously described [36,40]. Control locusts were injected with an equal concentration of *dsGFP*. Locusts (*n* = 15) were dissected into 5 pools per condition at N5WD to analyze the levels of *Schgr*-CCAPpre, *Schgr*-CCAPR-1, -2 and -3 transcripts.

**Rescue experiment.** Locusts were injected with 4 µL *dsCCAPpre* (100 ng/µL) within 24 h after the N4-N5 molt. Boost injections were given at N4D4, N5D1, N5D4 and N5D7. Control locusts were injected with 4 µL *dsGFP* (100 ng/µL). As of N5, *dsCCAPpre*-injected nymphs were additionally injected daily with 4 µL CCAP (0.92 pg/µL) in *S. gregaria* saline (1L: 8.766 g, NaCl; 0.188 g CaCl_2_; 0.746 g KCl; 0.407 g MgCl_2_;0.336 g NaHCO_3_; 30.807 g sucrose; 1.897 g trehalose; pH 7.2) in the rescue condition, or only with *S. gregaria* saline as a control. Likewise, control nymphs injected with *dsGFP* were additionally injected with *S. gregaria* saline starting at N5. 30 nymphs were observed per condition.

### 4.9. RNA Extraction, cDNA Synthesis and qRT-PCR

RNA extraction, cDNA synthesis and qRT-PCR were performed as previously described [36]. Primers used for qRT-PCR profiling are described in Appendix A.

### 4.10. In Vitro Incubation of Prothoracic Glands

PGs of N5IW nymphs were micro-dissected in *S. gregaria* saline solution and incubated in 50 µL Grace’s Insect medium (GI) (Invitrogen, Waltham, MA, USA) containing 1 µM CCAP or in GI without peptide. PG incubations were carried out in a 96-well culture plate sealed with a parafilm membrane, and maintained in a humidified incubator at 30 °C. After 24 h incubation, PG were removed, and the medium was transferred into 250 µL cold ethanol and extracted as described for the 20E enzyme immunoassay. As additional control, GI without PG was also incubated in a similar manner and subsequently extracted.

## 5. Conclusions

In conclusion, the presented data demonstrated an evolutionary conserved role of CCAP as a key regulator of the ecdysis motor program in the desert locust, a hemimetabolan species (Orthoptera, Polyneoptera) [17]. This finding suggests that this key role of CCAP was already present in the common ancestor of holometabolan and hemimetabolan insects. However, since the exact function of peptides in the ecdysteroid-induced neuropeptidergic cascade differs across species, it is possible to hypothesize that the ancestral regulatory network that controls ecdysis has incorporated several alterations during insect evolution [13,14].

In addition, we also revealed an intricate interaction between the ecdysteroid and CCAP signaling systems in locust post-embryonic development. Based on our data we conclude that in addition to its physiological significance in ecdysis, CCAP also behaves as a prothoracicostatic peptide suppressing ecdysteroidogenesis in the prothoracic gland at the end of the molt cycle. This is the first report on the prothoracicostatic activity of the CCAP signaling system, as well as on the first identified prothoracicostatic peptide in a hemimetabolan insect species [2,3].

Based on our results, no definite conclusion can be made on whether this prothoracicostatic activity of CCAP can be decoupled from its role in ecdysis induction. Further research will be needed to reveal the complex mechanistic network that consists of multiple functionally interacting signaling pathways. In addition, ecdysteroids are (cholesterol-) lipid-derived hormones and their regulation might perhaps be situated in a broader metabolic context. According to some literature reports [41,42], CCAP signaling in insects may indeed also be involved in metabolic regulation. Further research will be needed to address the novel and interesting questions arising from our study.

Given the phylogenetic position of *S. gregaria*, our findings represent an important addition to the current knowledge and understanding of the regulation of the molt cycle and subsequent ecdysis, vital processes for growth and development of insects, by far the most speciose class of organisms on our planet. Moreover, the lethal phenotype observed upon silencing essential genes involved in the control of ecdysis also reveals their potential as candidate targets for the development of novel pest management strategies.

## Figures and Tables

**Figure 1 ijms-22-13465-f001:**
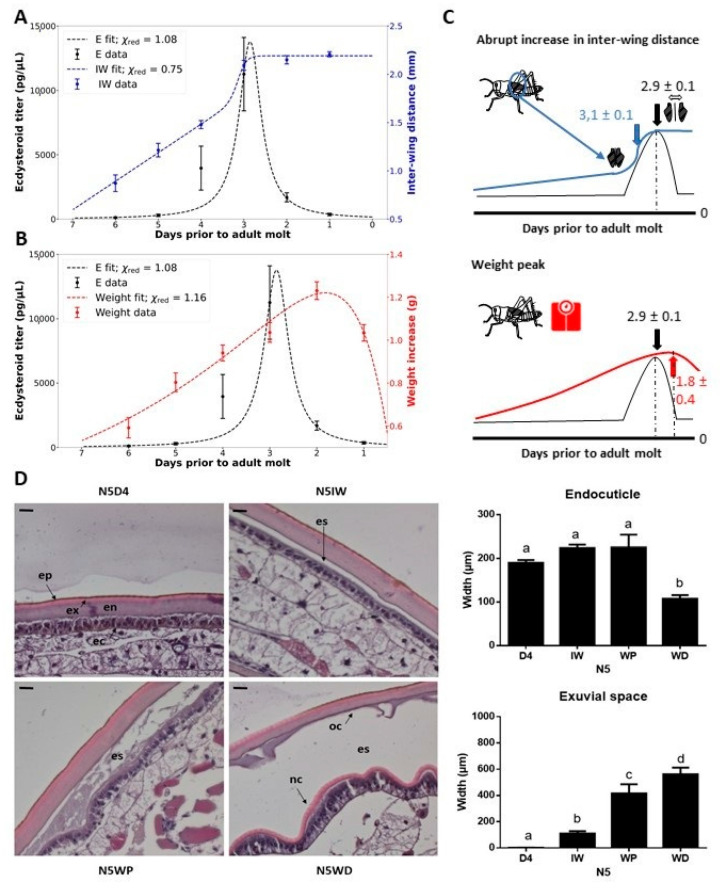
Fifth instar morphology in correlation to the ecdysteroid peak and structure of the integument of fifth instar (N5) nymphs. (**A**) Inter-wing distance (mm) (right *y*-axis) is displayed in blue and (**B**) body weight increase (g) (right *y*-axis) is shown in red, while the ecdysteroid titer (pg/µL) is displayed in black (left *y*-axis). The data represent the mean ± S.E.M. of 14 female nymphs followed throughout the fifth nymphal stage. The reduced Chi-square values (χ^2red^) of each fit are displayed. (**C**) Graphic representations of the temporal changes in physical characteristics that indicate developmental time points. Estimations of developmental time points by the fitted functions are displayed. (**D**) Molt-related changes in the morphology of the integument of *S. gregaria* N5 nymphs. Paraffin-embedded sections of the integument of the third abdominal tergite were stained with hematoxylin and eosin (H&E). Scale bar = 100 µm. Sampling points were N5D4, N5IW, N5WP and N5WD. The data represent the mean ± S.E.M. of 5 nymphs. Significant differences (*p* < 0.05) were calculated by One-way ANOVA. Abbreviations: N5 = fifth nymphal instar, D4 = day 4 of the fifth instar, IW = inter-wing distance of at least 1.8 mm, WP = weight peak, WD = weight decrease, ep = epicuticle, ex = exocuticle, en = endocuticle, ec = epidermal cells, oc = old cuticle, nc = new cuticle, es = exuvial space.

**Figure 2 ijms-22-13465-f002:**
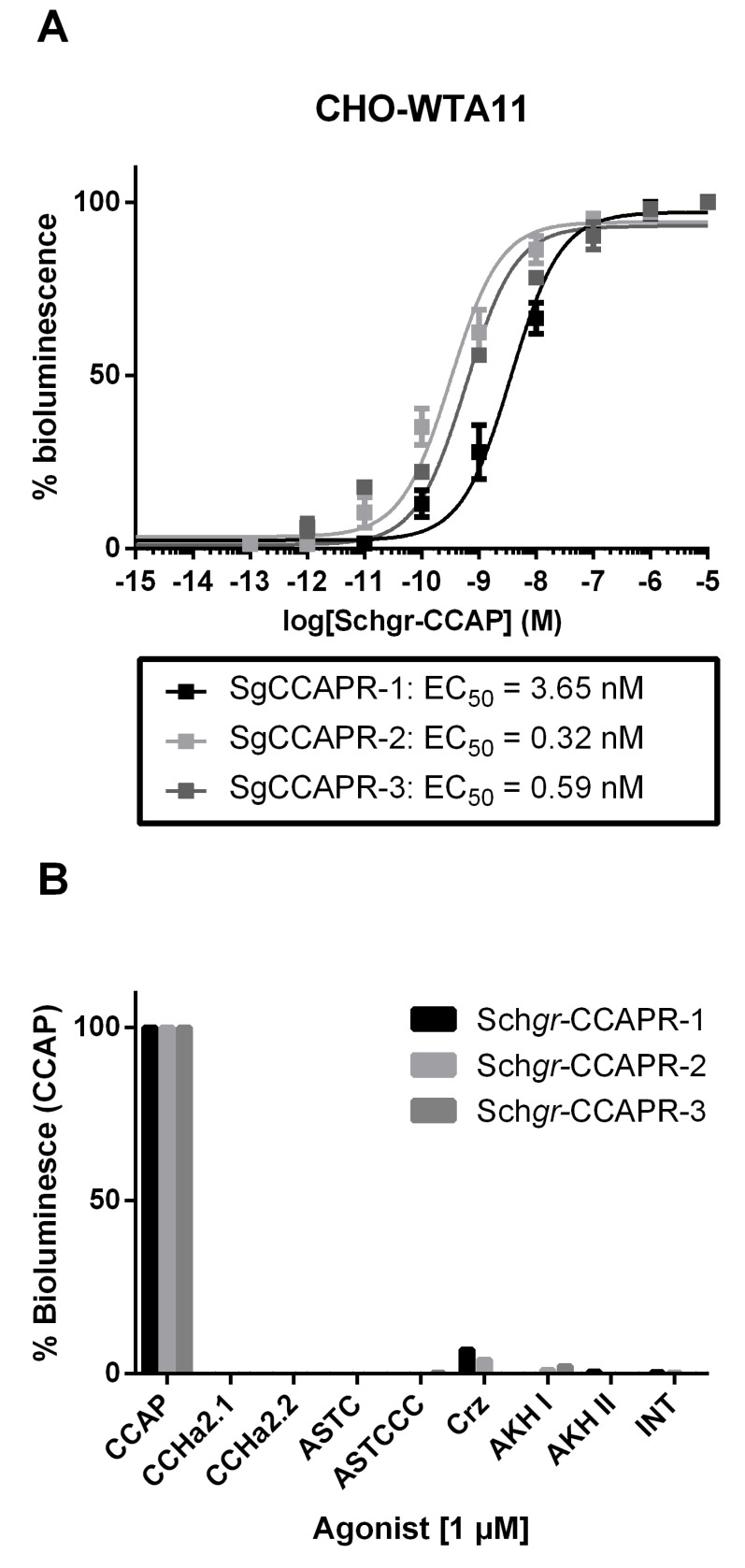
Aequorin-based bioluminescent responses in cell-based receptor activity assays. (**A**) Dose-response curves induced by CCAP in CHO-WTA11-*Schgr*-CCAPR-1, -2 or -3 cells. The EC_50_ values are indicated. (**B**) Results for aequorin-based bioluminescent response induced in CHO-WTA11-*Schgr*-CCAPR-1, -2 and -3 cells upon addition of 1 µM of peptide. These cell-based receptor activity data were obtained from three independent experiments. The cell-based assays were executed in three independent transfections. Error bars in dose-response curves represent the S.E.M and the 100% level refers to the maximal response level. Agonist selectivity was normalized to bioluminescent response acquired by exposure to CCAP. The zero-response level corresponds to treatment with BSA medium only. Abbreviations: *Schgr* = *Schistocerca gregaria*, CCAP = crustacean cardioactive peptide, CCAPR = CCAP receptor, CCHa = CCHamide, ASTC = allostatin C, ASTCCC = allostatin CCC, Crz = corazonin, AKH I(I) = adipokinetic hormone I(I), INT = inotocin.

**Figure 3 ijms-22-13465-f003:**
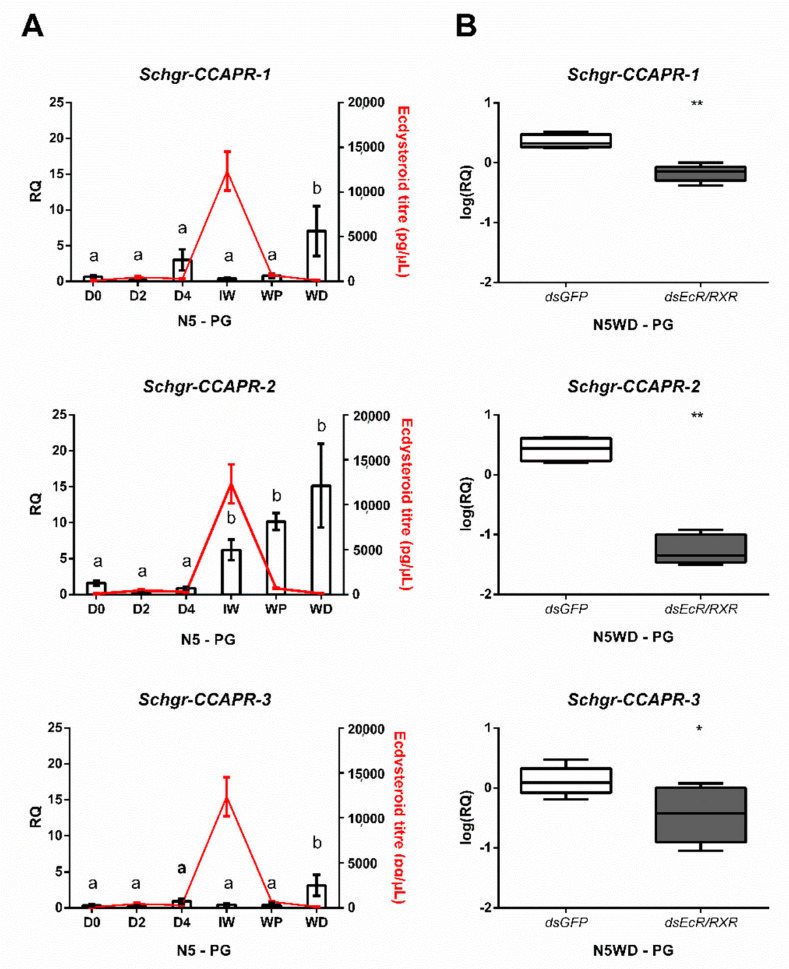
Expression of *Schgr*-CCAPR transcripts in the prothoracic glands. (**A**) Temporal expression profiles of *Schgr*-CCAPR-1, -2, and -3 in prothoracic glands (PG) throughout the fifth nymphal stage (N5) of *S. gregaria*. The data represent mean ± S.E.M. of five independent pools of three nymphs each, run in duplicate. Significant differences (*p* < 0.05) are indicated by distinct letters and were calculated by One-way ANOVA, followed by Dunnett’s test. Ecdysteroid titers (expressed in pg/µL, red line) were measured in hemolymph samples taken from these N5 nymphs. Data represent mean ± S.E.M. of five independent pools of three nymphs. (**B**) Effect of *Schgr*-EcR/RXR depletion on transcript levels of the CCAP receptors in the prothoracic glands (PG). Nymphs were injected at the day of the molt (D0) into the fifth nymphal stage (N5). A boost injection was administered at N5D3. The box plots (min to max) represent data obtained from 5 independent pools of three animals, run in duplicate. Significant differences (*p* < 0.05 and *p* < 0.01) are indicated by asterisks (* and **, respectively) and calculated by a two-sided unpaired *t*-test.

**Figure 4 ijms-22-13465-f004:**
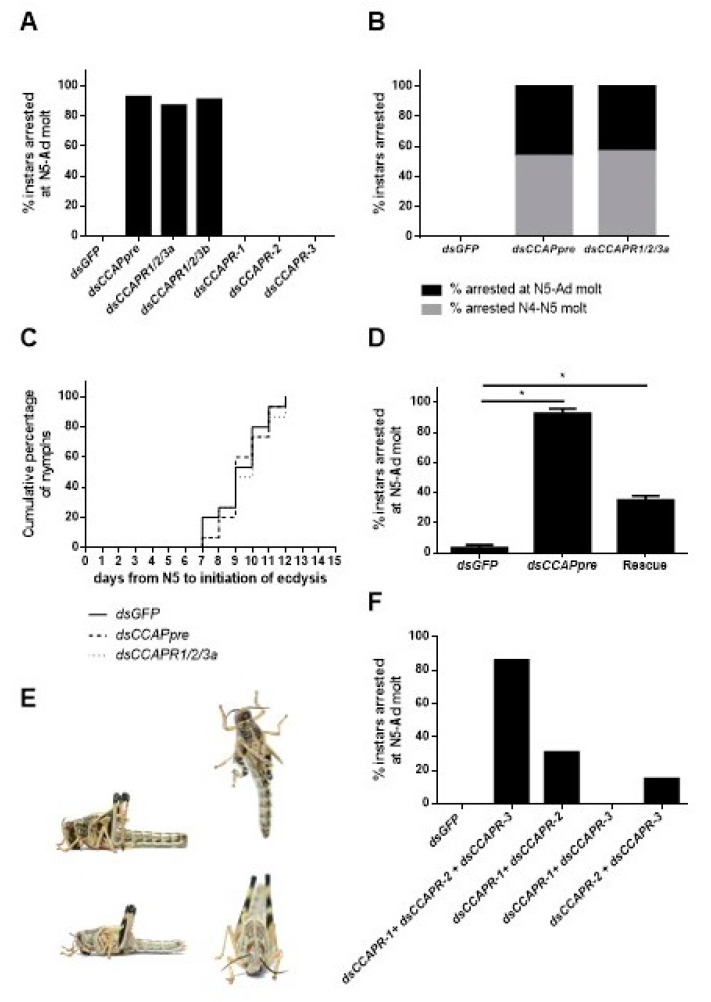
Phenotypic effect of *Schgr*-CCAPpre or *Schgr*-CCAPR-1, -2 and/or -3 depletion on ecdysis. (**A**) Percentage of nymphs displaying ecdysis arrest at nymphal-adult (N5-Ad) molt (*n* = 25, per construct). Nymphs were injected at N4D0, N4D4, N5D1, N5D4 and N5D7 with 400 ng of *dsGFP* (control), *dsCCAPpre*, *dsCCAPR1/2/3a*, *dsCCAPR1/2/3b*, *dsCCAPR-1*, *dsCCAPR-2* or *dsCCAPR-3*. (**B**) Percentage of nymphs displaying ecdysis arrest at nymphal-nymphal (N4–N5) and N5-Ad molts (*n* = 25, per construct). Nymphs were injected in late N3, N4D1 and N4D4 with 400 ng of *dsGFP* (control), *dsCCAPpre* or *dsCCAPR1/2/3a*. (**C**) The timing of ecdysis initiation in *dsGFP* (control), *dsCCAPpre* (*Schgr*-CCAPpre knockdown) and *dsCCAPR1/2/3a* (*Schgr*-CCAPRs knockdown) injected nymphs, as observed starting from freshly molted fifth instar nymphs. A log-rank (Mantel-Cox) test was used to check for significant differences in ecdysis initiation between populations. (**D**) Rescue of *Schgr*-CCAPpre depleted nymphs (*n* = 30) by additional daily injections of 0.92 pg CCAP in 4 µL *S. gregaria* saline in N5. In the control groups, nymphs received *dsGFP* injections or *dsCCAPpre* injections (*n* = 31), with additional daily injections of *S. gregaria* saline in N5. Significant differences (*p* < 0.05) are indicated by an asterisk (*) and calculated by the Fisher’s exact test. (**E**) Ecdysis deficient phenotype (top photos). New (adult) cuticle is visible after manual removal of old nymphal cuticle (bottom photos). (**F**) Effect caused by injections of multiple selective *dsCCAPR* constructs (two or three) on ecdysis. Nymphs were injected with combinations of *dsCCAPR* constructs (400 ng/4 µL per construct) in day 0 and day 3 in the fourth instar, as well as on day 1, day 4 and day 7 in the fifth instar. Different combination groups of *dsRNA* constructs: *dsGFP* (*n* = 36, control), *dsCCAPR-1* + *dsCCAPR-2* + *dsCCAPR-3* (*n* = 52), *dsCCAPR-1* + *dsCCAPR-2* (*n* = 48), *dsCCAPR-1* + *dsCCAPR-3* (*n* = 37) and *dsCCAPR-2* + *dsCCAPR-3* (*n* = 29).

**Figure 5 ijms-22-13465-f005:**
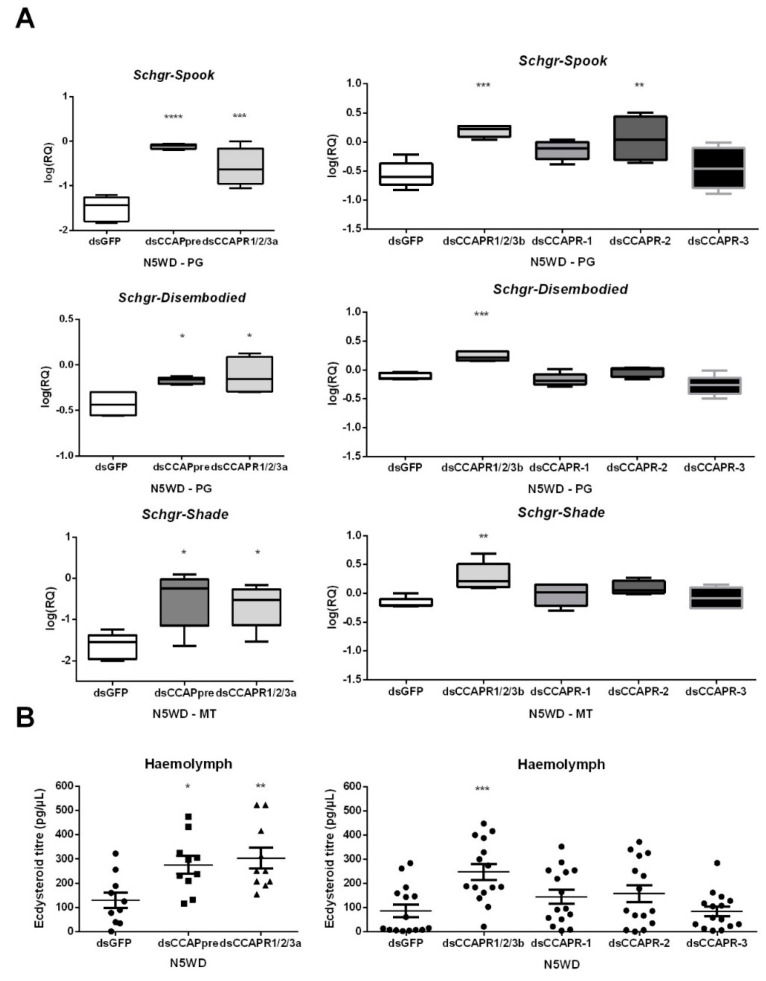
Effects of CCAP(pre/R) depletion on (**A**) expression of *Halloween* genes and (**B**) ecdysteroid levels in locusts of the fifth nymphal stage (N5) at weight decrease (WD). The box plots (min to max) in panel A represent data obtained from 5 independent pools of three locusts, run in duplicate. Ecdysteroid titres in panel B indicate the mean ± S.E.M. (*n* ≥ 10). Significant differences (*p* < 0.05, *p* < 0.01, *p* < 0.001 and *p* < 0.0001) are indicated by asterisks (*, **, *** and ****, respectively) and were calculated by One-way ANOVA, followed by a Dunnett’s Test, on log-transformed data. Abbreviations: PG = prothoracic gland, MT = Malpighian tubules, CCAPpre = crustacean cardioactive peptide precursor, CCAPR = CCAP receptor, *Schgr* = *Schistocerca gregaria*.

**Figure 6 ijms-22-13465-f006:**
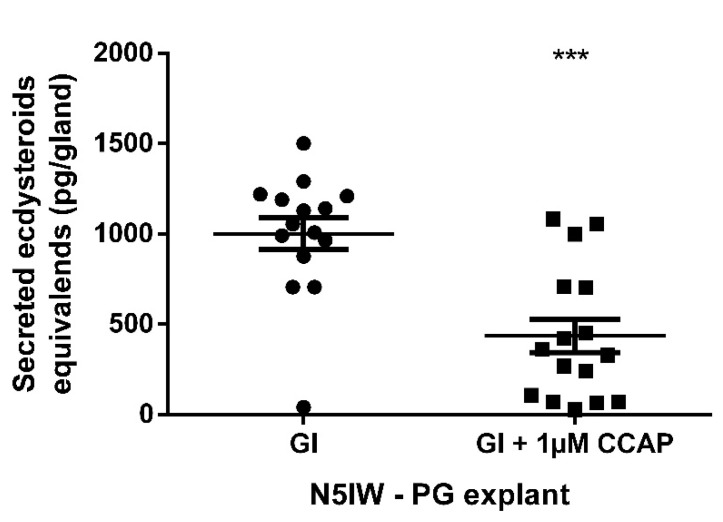
Ecdysteroid secretion by prothoracic gland (PG) explants incubated for 24 h in a volume of 50 µL Grace’s insect medium (GI) with or without 1 µM CCAP. PGs were micro-dissected from fifth instar (N5) nymphs at N5IW (inter-wing distance of at least 1.8 mm; GI, *n* = 15; GI + 1 µM CCAP, *n* = 16) Each data point represents 20E-equivalents secreted by PG from a single nymph. Mean ± S.E.M. is also indicated per group. Significant difference (*p* < 0.001) is indicated by asterisks (***) and was calculated by an unpaired *t*-test with Welch’s correction. Used abbreviations: CCAP = crustacean cardioactive peptide and GI = Grace’s Insect Medium.

## Data Availability

All data generated or analyzed during this study are included in this published article; its datasets are available from the corresponding author upon reasonable request. Protein sequence accessions: The *S. gregaria* CCAP(pre/R) sequences were deposited in GenBank at the National Center for Biotechnology Information (NCBI): *Schgr*-CCAPpre (GenBank acc No MZ647656), *Schgr*-CCAPR-1 (GenBank acc No MZ647657), *Schgr*-CCAPR-2 (GenBank acc No MZ647658) and *Schgr*-CCAPR-3 (Genbank acc No MZ647659).

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
