# Peer review of "Prothoracicostatic Activity of the Ecdysis-Regulating Neuropeptide Crustacean Cardioactive Peptide (CCAP) in the Desert Locust"

_ijms, 2021, doi:10.3390/ijms222413465_

Round 1

Reviewer 1 Report

This an interesting manuscript describing identification and functional characterization of CCAP precursor and its receptors in the desert locust, Schistocerca gregaria. The authors showed for the first time prothoracicostatic activity of CCAP and determined that knockdown of CCAP or its receptors led to increased transcript levels of specific Halloween genes. I only have several minor comments and suggestions:

Introduction, page 2, line 54: PSTPs were originally named myoinhibitory peptides, so please, include “or myoinhibitory peptides (MIPs)“

Results: The authors should first describe identification and characterization of CCAPRs and then show their expression profiles in the prothoracic glands. I suggest to fuse chapters 2.2 - 2.3 and switch Figs 2 and 3.

Expression of Schgr-CCAPR transcripts in prothoracic glands and other tissues (page 4, lines 148-163) should be described in a separate chapter 2.3 and shown in Fig. 3.

Page 7, line 219: It would be more appropriate to write: “...transcripts were reduced by RNAi to 14% and 39%...”

Page 7, line 252: It is difficult to distinguish different experimental groups (dsCCAPR1/2/3a, dsCCAPR1/2/3b and dsCCAPR1/2/3) so it would be good to write “...control nymphs injected with dsGFP or dsCCAPR-1,2,3 that target highly diverging regions of respective receptors ...“

Page 8, line 258: please change Figure 3A to 4A.

Page 10, line 313: please remove “that were”

Page 13, line 416: please replace "S. gregaria" with "this species"

Author Response

We thank the editor and the reviewers for their constructive remarks on our original manuscript. These helped us to revise our manuscript accordingly.

Please find below a point-by-point reply to the different comments, as well as a description of the changes we made during our manuscript revision. For the editor’s and reviewers’ convenience, we also highlighted our changes within the manuscript.

Reviewer #1: This an interesting manuscript describing identification and functional characterization of CCAP precursor and its receptors in the desert locust, Schistocerca gregaria. The authors showed for the first time prothoracicostatic activity of CCAP and determined that knockdown of CCAP or its receptors led to increased transcript levels of specific Halloween genes. I only have several minor comments and suggestions:

We thank the reviewer for the positive comments which are very helpful to improve our manuscript.

Introduction, page 2, line 54: PSTPs were originally named myoinhibitory peptides, so please, include “or myoinhibitory peptides (MIPs)“

We have now included the reviewer’s suggestion in the Introduction Section Line 54.

Results: The authors should first describe identification and characterization of CCAPRs and then show their expression profiles in the prothoracic glands. I suggest to fuse chapters 2.2 - 2.3 and switch Figs 2 and 3.

We have followed the reviewer’s text structure suggestion and fused the original Results Sections 2.2. and 2.3, which is now titled “Identification, molecular and functional characterization of the S. gregaria CCAP precursor and receptors” (Lines 123-124). Figure 3 is now Figure 2 (Lines 155,180) and vice versa (Lines 203,206,226). In addition, Figure S5 is now Figure S4 (Lines 161,165) and vice versa (Lines 189, 193, 196, 198). This has also been adapted in the Supplementary Section: Lines 648-652.

Expression of Schgr-CCAPR transcripts in prothoracic glands and other tissues (page 4, lines 148-163) should be described in a separate chapter 2.3 and shown in Fig. 3.

We have followed the reviewer’s text structure suggestion and made a new Results Section 2.3 entitled “Localization and developmental expression patterns of the S. gregaria CCAP precursor and receptors” (Lines 186-187).

Page 7, line 219: It would be more appropriate to write: “...transcripts were reduced by RNAi to 14% and 39%...”

We have adapted our text in accordance with the reviewer’s suggestion (Line 224).

Page 7, line 252: It is difficult to distinguish different experimental groups (dsCCAPR1/2/3a, dsCCAPR1/2/3b and dsCCAPR1/2/3) so it would be good to write “...control nymphs injected with dsGFP or dsCCAPR-1,2,3 that target highly diverging regions of respective receptors ...“

The sentence has been changed to “By contrast, nymphs injected with dsGFP (control) or dsCCAPR-1, -2 or -3 (targeting highly diverging regions of the respective receptors) managed to shed their old cuticle” in Results Section 2.5.2. (Line 256-258).

Page 8, line 258: please change Figure 3A to 4A.

We apologize for this typo and corrected this in Results Section 2.5.2. (Line 263).

Page 10, line 313: please remove “that were”

We have followed the reviewer’s suggestion and removed “that were” from the sentence in Results section 2.6. Line 319.

Page 13, line 416: please replace "S. gregaria" with "this species"

We have followed the reviewer’s suggestion and replaced “S. gregaria” with “this species” in Discussion Section 3.4. Line 421.

Reviewer 2 Report

Reviewer’s comment                                                                                       

I have carefully reviewed the manuscript " Prothoracicostatic activity of the Ecdysis-regulating Neuropeptide Crustacean Cardioactive Peptide (CCAP) in the Desert Locust ". The manuscript is well written with a balanced Introduction and subsequent Discussion based on the results, which I overall appreciate.

The experiments are well planned and well performed. The methods are a mixture of traditional assays but include modern molecular biological approaches as RNAi‘s. The manuscript suggests CCAP signaling system not only controls the ecdysis motor program in the Desert Locust but also plays a prothoracicostatic role.

Moreover, the most important observation of lethal ecdysis deficient phenotype occurrence after essential ecdysis regulating genes silencing promotes CCAP as a potential candidate to be further manipulated in novel pest management strategies. This study has the potential to open a new gateway in controlling this destructive pest worldwide and to save food crops of economic benefit.

Yet, therein are a few questions, which I would appreciate to be addressed

  1. What is the main reason to select two parameters such as inter-wing distance and body weight?
  2. What factors are taken into consideration while taking these measurements?
  3. Line 574-589: In the case of Schgr-EcR/Schgr-RXR knockdown you use 50ng/ul and in the other two cases you used 100ng/ul. Can you provide the explanation? It will be helpful for the readers.
  4. How about lipid accumulation, do you try some things with this orientation?
  5. How well this concept works on pest control strategies? The total study was taken place on nymph stage 5. How about the early stages? Does this application alters the total physiology of the insect and is it stage and sex-specific? Adding a few points to the conclusion will make it more interesting.

Conclusively, this is a very well-conceived and written paper. With a variety of techniques, the authors have uncovered an unknown function of CCAP, which not only participates in the ecdysis motor program but also exhibits prothoracicostatic role. For this reason, I suggest this Manuscript be accepted and published with minor revisions after clarification/incorporations of my questions and suggestions.

Some of the observed typos need to be corrected.

Line: 54 Bommo-myosuppressin : Bommo should not be in Italics.

Line 75 and 78: It should be a desert locust. Please correct it throughout the manuscript (Section: Conclusion).

Line 244: Figure 3B mislabelled. I believe it is Figure 4B.

Line 251: figure 3E bottom doesn't exist. It is again mislabelled. It should be Figure 4E.

Line 281: (E) is mislabelled as (C). Please correct it.

Line 258: Figure mislabelling (Figure 4A not 3A) ( please check it throughout the text and correct).

Line 281: Please correct the mislabelling.

Line 288: Halloween needs to be in Italics.

Author Response

We thank the editor and the reviewers for their constructive remarks on our original manuscript. These helped us to revise our manuscript accordingly.

Please find below a point-by-point reply to the different comments, as well as a description of the changes we made during our manuscript revision. For the editor’s and reviewers’ convenience, we also highlighted our changes within the manuscript.

Reviewer #2: I have carefully reviewed the manuscript "Prothoracicostatic activity of the Ecdysis-regulating Neuropeptide Crustacean Cardioactive Peptide (CCAP) in the Desert Locust ". The manuscript is well written with a balanced Introduction and subsequent Discussion based on the results, which I overall appreciate.

The experiments are well planned and well performed. The methods are a mixture of traditional assays but include modern molecular biological approaches as RNAi‘s. The manuscript suggests CCAP signaling system not only controls the ecdysis motor program in the Desert Locust but also plays a prothoracicostatic role.

Moreover, the most important observation of lethal ecdysis deficient phenotype occurrence after essential ecdysis regulating genes silencing promotes CCAP as a potential candidate to be further manipulated in novel pest management strategies. This study has the potential to open a new gateway in controlling this destructive pest worldwide and to save food crops of economic benefit.

Conclusively, this is a very well-conceived and written paper. With a variety of techniques, the authors have uncovered an unknown function of CCAP, which not only participates in the ecdysis motor program but also exhibits prothoracicostatic role. For this reason, I suggest this Manuscript be accepted and published with minor revisions after clarification/incorporations of my questions and suggestions.

We are very glad to read that the reviewer considers our manuscript as an overall well-written and structured piece of work that has novelty and the potential to promote CCAP as a candidate for pest-management strategies. We are also very grateful to the reviewer for his/her analysis of the text, and for the constructive questions and remarks that helped us improving our manuscript.

Yet, therein are a few questions, which I would appreciate to be addressed
What is the main reason to select two parameters such as inter-wing distance and body weight? What factors are taken into consideration while taking these measurements?

Response: To standardize sampling at distinct moments in the molting cycle, we indeed correlated two physical indicators of fifth instar (N5) nymphs to the ecdysteroid titer that is considered as a prerequisite to induce the molting process. These physical characteristics were selected according to their suitability to be applied as indicators of distinct molt cycle moments during experiments. Therefore, only easily, directly and non-invasively measurable, consistently changing, physical characteristics were considered. To clarify this in the text, we added a sentence to the Materials and Methods Section, paragraph 4.2. “These anatomical indicators were selected based on their applicability in experimental set-ups.” (Lines 449-450).

Line 574-589: In the case of Schgr-EcR/Schgr-RXR knockdown you use 50 ng/µl and in the other two cases you used 100 ng/µl. Can you provide the explanation? It will be helpful for the readers.

Response: In locusts, RNA interference is a very robust process that doesn’t require injection of very large quantities of dsRNA in order to be highly effective. “The injected dsRNA quantities were based on pilot experiments, in which the different dsRNAs were tested, since the optimal dose may vary depending on the targeted gene transcript.” Determining the minimal concentration that is still very effective is also the most cost-effective research strategy in the long run. For the dsEcR and dsRXR constructs this was 50 ng/µL (Lenaerts et al., 2016 doi: 10.1016/j.ibmb.2016.05.003), while for the dsCCAP(pre/R) constructs we found this to be 100 ng/µL.  The underlined sentence has been added to Materials and Methods Section, paragraph 4.8. (Lines 582-584).

How about lipid accumulation, do you try some things with this orientation?

Response: We thank the reviewer for this interesting question. Ecdysteroids are (cholesterol-) lipid-derived hormones and their regulation might perhaps be situated in a broader metabolic context. According to some literature reports, CCAP signaling in insects may indeed also be involved in metabolic regulation. And nearly 25 years ago, our lab found that CCAP can induce the release of Adipokinetic Hormone (AKH) from corpora cardiaca of locusts (Veelaert et al., 1997 doi: 10.1210/endo.138.1.4855). Interestingly, in the current study we now describe the occurrence of CCAPR-encoding transcripts in these neuroendocrine organs. On the other hand, we noticed in a preliminary study of our lab that AKH knockdown didn’t affect the ecdysis process. This may suggest that, if accumulation or mobilization of lipids would be crucial for this process, their regulation probably does not exclusively rely on AKH. Further research will be needed to address this question.

How well this concept works on pest control strategies?

Response: It is clear that injection of dsRNA to knock down Schgr-CCAP(pre/R) transcripts, as shown in our manuscript, severely disrupts ecdysis and strongly affects the locusts’ survival chances. This indeed makes the Schgr-CCAPRs promising candidate targets for development of novel insecticides. The same is probably true for some other regulators involved in this process.

Even though locusts have a very robust systemic RNAi response, delivery of ‘naked dsRNA’ via the oral route was shown to be ineffective, which is -at least in part- due to the very high dsRNase activity in the gut lumen (Luo et al., 2013 doi: 10.1111/imb.12046; Spit et al., 2017 doi: 10.1016/j.ibmb.2017.01.004; Vogel et al., 2019 doi: 10.3389/fphys.2018.01912). Therefore, several delivery systems, such as nanoparticles or microbial systems, are being studied that may more effectively enable protection (against degradation) and uptake of dsRNA from the gut lumen. So, further research is needed for optimizing dsRNA delivery.

Another possible approach could be based on high-throughput small compound screening, and/or on biorational design strategies, to identify novel molecules that can specifically target crucial insect hormone/neuropeptide receptors, such as the Schgr-CCAPRs. Some mimetic neuropeptide analogs with strongly increased biostability have already been shown to have insecticidal activities (Nachman et al.,2012, doi: 10.1016/j.peptides.2011.11.009; Down et al., 2010, doi: 10.1016/j.peptides.2009.06.017).

The total study was taken place on nymph stage 5. How about the early stages?

Response: As was stated in Lines 246-249: “Moreover, injections that started from late third instar nymphs, with these three constructs, led to 100% ecdysis deficiency, considering the combined effect on the N4-N5 molt and the N5-Ad molt in the tested population” and as shown in Figure 4B, injections that started in N3 stage also led to ecdysis defects during the N4-N5 transition in a large part of the tested locust population. So, we expect that CCAP is also needed during nymphal-to-nymphal transitions (in line with its conserved role in inducing the ecdysis motor program).

Does this application alters the total physiology of the insect and is it stage and sex-specific?

Response: We have checked whether our dsRNA treatments affected the timing of the molting cycle associated physical characteristics during the N5 stage, but this was not the case. Also, the locusts appeared very normal until the expected starting moment of their ecdysis behavior (which was arrested in case of CCAP knockdown). We have indeed focused on the N5 stage, because the physical indicators used for sampling were correlated to the ecdysteroid titer during this stage. This certainly does not imply that CCAP only acts in the N5 stage (see also our comment above). The observed effects are not sex-specific, since in preliminary experiments both male and female locusts displayed the same ecdysis-deficient phenotype upon knockdown of CCAP. We performed our study on females, because we wanted to reduce variability as the physical indicators were initially standardized for females. However, we observed (later) that these indicators can also be utilized for male locusts (while the average weight of a male locust is lower than that of a female, males also reach a weight peak, followed by a weight drop).

Adding a few points to the conclusion will make it more interesting.

Response: We prefer to keep the attention of the reader well-focused on the main scientific conclusions from our experimental findings: (1) CCAP is a central regulator of ecdysis in the hemimetabolan species, S. gregaria and (2) CCAP also acts as a prothoracicostatic peptide. We addressed the different points made by Reviewer #2 in our responses above.

In our revised manuscript (adapted based on all reviewers’ comments), the Conclusion section now reads as follows: “In conclusion, the presented data demonstrated an evolutionary conserved role of CCAP as a key regulator of the ecdysis motor program in the desert locust, a hemimetabolan species (Orthoptera, Polyneoptera). This finding suggests that this key role of CCAP was already present in the common ancestor of holometabolan and hemimetabolan insects. However, since the exact function of peptides in the ecdysteroid-induced neuropeptidergic cascade differs across species, it is possible to hypothesize that the ancestral regulatory network that controls ecdysis has incorporated several alterations during insect evolution (White & Ewer 2014 doi: 10.1146/annurev-ento-011613-162028; Gammie and Truman 1997 doi: 10.1523/JNEUROSCI.17-11-04389 ).

In addition, we also revealed an intricate interaction between the ecdysteroid and CCAP signaling systems in locust post-embryonic development. Based on our data we conclude that in addition to its physiological significance in ecdysis, CCAP also behaves as a prothoracicostatic peptide suppressing ecdysteroidogenesis in the prothoracic gland at the end of the molt cycle. This is the first report on the prothoracicostatic activity of the CCAP signaling system, as well as on the first identified prothoracicostatic peptide in a hemimetabolan insect species (Pan et al., 2021 doi: 10.1016/j.cois.2020.09.004; Smith et al., 2010 ISBN 9780123847492).

Based on our results, no definite conclusion can be made on whether this prothoracicostatic activity of CCAP can be decoupled from its role in ecdysis induction. Further research will be needed to reveal the complex mechanistic network that consists of multiple functionally interacting signaling pathways. Ecdysteroids are (cholesterol-) lipid-derived hormones and their regulation might perhaps be situated in a broader metabolic context. According to some literature reports (Mikani et al., 2015 doi: 10.1007/s00441-015-2242-4; Williams et al., 2020 doi: 10.1073/pnas.1914037117), CCAP signaling in insects may indeed also be involved in metabolic regulation. Further research will be needed to address the novel and interesting questions arising from our study.

Given the phylogenetic position of S. gregaria, our findings represent an important addition to the current knowledge and understanding of the regulation of the molt cycle and subsequent ecdysis, vital processes for growth and development of insects, by far the most speciose class of organisms on our planet. Moreover, the lethal phenotype observed upon silencing essential genes involved in the control of ecdysis also reveals their potential as candidate targets for the development of novel pest management strategies.” (Lines 615-644).

Some of the observed typos that needed to be corrected:

Line: 54 Bommo-myosuppressin : Bommo should not be in Italics.

Line 54: We have corrected our mistake

Line 75 and 78: It should be a desert locust. Please correct it throughout the manuscript (Section: Conclusion).

Line 21, 75, 78, 617: We have removed the capital letters for “desert locust”.

Line 244: Figure 3B mislabeled. I believe it is Figure 4B.

Line 249: We apologize for this typo. Indeed it must be Figure 4B. Our mistake has now been corrected.

Line 251: figure 3E bottom doesn't exist. It is again mislabeled. It should be Figure 4E.

Line 256: We apologize for this typo. Indeed, it must be Figure 4E. Our mistake has now been corrected.

Line 281: (E) is mislabeled as (C). Please correct it.

Line 287 (Legend Figure 4): We apologize for this typo. Indeed, it must be E. Our mistake has now been corrected.

Line 258: Figure mislabeling (Figure 4A not 3A) (please check it throughout the text and correct).

Line 264: We apologize for this typo. Indeed, it must be 4A. Our mistake has now been corrected.

Line 288: Halloween needs to be in Italics.

Line 294: We have corrected our mistake.

Reviewer 3 Report

The manuscript by Verbakel et al., examines the function of Crustacean Cardioactive Peptide in the Desert Locust  Schistocerea gregraria.  Previous studies on this neuropeptide signaling system have demonstrated a role in control ecdysis motor programs in both hemi and holometabolan insects, but no clear role in regulation of ecdysteroid pulse generation that initiates molting.  The authors begin their study by carefully various morphological features  with the timing of the fifth instar ecdysteroid peak and the transition to the adult form.    This gives them a point of reference for examining the potential roles of CCAP signaling in regulation of ecdysteroid production at the fifth instar to adult transition. They then clone the CCAP gene as well as its three putative receptors and examine their temporal and spatial expression patterns.  Interestingly, they observe that all three receptors show various levels of expression in the prothoracic gland which are regulated by Ecdysone signaling since they are reduced through RNAi mediated Ecdysone Receptor knockdown.  Next they carry out cell culture signaling assays to demonstrate that each putative receptor responds to nanomolar levels of schg-CCAP. The authors then carry out the key set of experiments which is to examine the consequence on 5 instar to adult molt  when either CCAP or its receptors is knockdown.  They find that RNAi mediated knockdown of either CCAP itself or the combination of all three receptors arrests  greater than 90% fail to complete ecdysis of the fifth instar to the Adult stage.  They then go on to show that knockdown of this signaling system leads to an increased ecdysone titer during the fifth instar stage likely due to increased levels of ecdysone biosynthetic enzyme transcript levels.  Lastly they demonstrate that addition of CCAP to isolated PG glands leads to decreased synthesis/release of ecdysteroid from in vitro cultured prothoracic glands.  They come to the novel and interesting conclusion that CCAP acts as a prothoracicostatic factor in S. Gregraria.

This is a thorough, well designed, and carefully executed study with some interesting conclusions.   The only comment I have is that the authors do not really make clear whether they what the relationship is between the abnormally high E titer in the knockdown animals and the failure to properly execute the ecdysis motor program.   Are the two connected, ie is the abnormally high E titer interfering with the execution of the motor program, or is the abnormal E titer inconsequential and the arrest is simply reflecting the direct requirement of CCAP signaling for initiating the motor program.   Would it be possible to examine this issue by injecting or feeding fifth instar nymphs E to increase its concentration  independent of CCAP manipulation to see if this blocks the transition.  If this is observed then it argues that the increased E titer is  actually detrimental for a proper developmental transition to occur. If this experiment is not possible, then a little more discussion on this point in the conclusions is warranted.

Minor point: lines 244, 251 and 258 refer to Figures 3B, 3E and 3A respectively. The actual Figure citations should  be 4B, 4E and 4A.

Author Response

We thank the editor and the reviewers for their constructive remarks on our original manuscript. These helped us to revise our manuscript accordingly.

Please find below a point-by-point reply to the different comments, as well as a description of the changes we made during our manuscript revision. For the editor’s and reviewers’ convenience, we also highlighted our changes within the manuscript.

Reviewer #3: The manuscript by Verbakel et al., examines the function of Crustacean Cardioactive Peptide in the Desert Locust Schistocerca gregraria.  Previous studies on this neuropeptide signaling system have demonstrated a role in control ecdysis motor programs in both hemi- and holometabolan insects, but no clear role in regulation of ecdysteroid pulse generation that initiates molting.  The authors begin their study by carefully various morphological features  with the timing of the fifth instar ecdysteroid peak and the transition to the adult form.    This gives them a point of reference for examining the potential roles of CCAP signaling in regulation of ecdysteroid production at the fifth instar to adult transition. They then clone the CCAP gene as well as its three putative receptors and examine their temporal and spatial expression patterns.  Interestingly, they observe that all three receptors show various levels of expression in the prothoracic gland which are regulated by Ecdysone signaling since they are reduced through RNAi mediated Ecdysone Receptor knockdown.  Next they carry out cell culture signaling assays to demonstrate that each putative receptor responds to nanomolar levels of Schgr-CCAP. The authors then carry out the key set of experiments which is to examine the consequence on 5 instar to adult molt  when either CCAP or its receptors is knockdown.  They find that RNAi mediated knockdown of either CCAP itself or the combination of all three receptors arrests  greater than 90% fail to complete ecdysis of the fifth instar to the Adult stage.  They then go on to show that knockdown of this signaling system leads to an increased ecdysone titer during the fifth instar stage likely due to increased levels of ecdysone biosynthetic enzyme transcript levels.  Lastly they demonstrate that addition of CCAP to isolated PG glands leads to decreased synthesis/release of ecdysteroid from in vitro cultured prothoracic glands.  They come to the novel and interesting conclusion that CCAP acts as a prothoracicostatic factor in S. gregraria.

This is a thorough, well designed, and carefully executed study with some interesting conclusions.  The only comment I have is that the authors do not really make clear whether they what the relationship is between the abnormally high E titer in the knockdown animals and the failure to properly execute the ecdysis motor program.   Are the two connected, i.e. is the abnormally high E titer interfering with the execution of the motor program, or is the abnormal E titer inconsequential and the arrest is simply reflecting the direct requirement of CCAP signaling for initiating the motor program.  Would it be possible to examine this issue by injecting or feeding fifth instar nymphs E to increase its concentration  independent of CCAP manipulation to see if this blocks the transition.  If this is observed then it argues that the increased E titer is  actually detrimental for a proper developmental transition to occur. If this experiment is not possible, then a little more discussion on this point in the conclusions is warranted.

Response: We are very glad to read that the reviewer considers the content of our paper as an overall well written and structured, well-designed study that has novelty and interesting conclusions. We are also very grateful to the reviewer for his/her analysis of the text, and her/his interesting comment that we have now addressed in our Conclusion section: “Based on our results, no definite conclusion can be made on whether this prothoracicostatic activity of CCAP can be decoupled from its role in ecdysis induction. Further research will be needed to reveal the complex mechanistic network that consists of multiple functionally interacting signaling pathways. ” (L631-634).  

We do not yet have clear evidence to fully exclude either of these hypotheses. It may indeed prove difficult to effectively decouple the ecdysis induction and prothoracicostatic effects of CCAP by experimental evidence. Given the systemic nature of the RNAi process in locusts, it is currently not possible to conditionally interfere with the spatio-temporal expression patterns of particular genes (in specific locations or at concrete developmental time points). Feeding or injecting ecdysteroids in locusts is known to induce rapid inactivation and/or clearance of excess hormone. Sustained (non-declining) ecdysteroid levels might interfere with the release of ecdysis triggering hormone (ETH, the central initiator of the entire ecdysis sequence) (Zitnan et al., 2012 doi: 10.1016/B978-0-12-384749-2.10007-X). ETH is known to act on CCAP producing neurons, from which CCAP is released to induce the ecdysis motor program (Kim et al., 2006 doi: 10.1073/pnas.0603459103). It might also be possible to inject CCAP (in addition to 20E) at different time points in control or CCAPpre RNAi knockdown animals (Figure 4D), to observe whether this would affect their ecdysis. However, injected peptide may not (efficiently) reach all target sites, considering the presence of peptidases, as well as the blood-brain barrier. On the other hand, it is also possible to reduce ecdysteroid synthesis by knocking down Halloween gene transcripts. Our lab published on this in the past (Marchal et al. 2011 doi: 10.1016/j.jinsphys.2011.05.009; Marchal et al. 2012 doi: 10.1016/j.jinsphys.2012.03.013). Although this treatment proved to effectively reduce the circulating ecdysteroid titers, it was not sufficient to disturb the molting process.

Therefore, additional experiments would be required to analyze whether the regulatory hierarchies described for some holometabolan research models are identical or very similar in locusts. The complex functional interactions between the different factors in this regulatory network, which probably also involve several feedback mechanisms, may make it very difficult to identify all contributions of one specific factor such as CCAP at a high spatio-temporal resolution. Further research will be needed to address this question by testing different hypotheses.

Minor point: lines 244, 251 and 258 refer to Figures 3B, 3E and 3A respectively. The actual Figure citations should  be 4B, 4E and 4A.

We apologize for these typos, they were corrected in the revised manuscript.